# A Truly Constant-time Distribution-aware Negative Sampling

## Abstract

Softmax classifiers with a very large number of classes naturally occur in many applications such as natural language processing and information retrieval. The calculation of full-softmax is very expensive from the computational and energy perspective. There have been a variety of sampling approaches to overcome this challenge, popularly known as negative sampling (NS). Ideally, NS should sample negative classes from a distribution that is dependent on the input data, the current parameters, and the correct positive class. Unfortunately, due to the dynamically updated parameters and data samples, there does not exist any sampling scheme that is truly adaptive and also samples the negative classes in constant time every iteration. Therefore, alternative heuristics like random sampling, static frequency-based sampling, or learning-based biased sampling, which primarily trade either the sampling cost or the adaptivity of samples per iteration, are adopted. In this paper, we show a class of distribution where the sampling scheme is truly adaptive and provably generates negative samples in constant time. Our implementation in C++ on commodity CPU is significantly faster, in terms of wall clock time, compared to the most optimized TensorFlow implementations of standard softmax or other sampling approaches on modern GPUs (V100s).

## 1 Introduction

Neural Networks (NN) have successfully pushed the boundaries of many application tasks, such as image or text classification (Wang et al., 2017; Yao et al., 2019), speech recognition (Dong et al., 2018) and recommendation systems (Zhang et al., 2015; Medini et al., 2019). Many hard AI problems are currently modeled as massive multiclass or multilabel problems leading to a drastic improvement over prior work. For example, popular NLP models predicts the best word, given the full context observed so far. Such models are becoming the state-of-the-art. Recommendation systems and related Information Retrieval (IR) problems are classical examples of machine learning with outrageously large outputs (Medini et al., 2019; Jain et al., 2019). In IR, given the user query, the task is to predict few relevant documents (or products) from among hundreds of millions possible documents, a typical machine learning problem with massive output space.

Owing to the significance of the problem, *machine learning with large output space* or alternatively also known as *extreme classification* is a field in itself (Bengio et al., 2019). A large number of classes naturally brings a new set of computational and memory challenge.

Fortunately, with access to our powerful Graphic Processing Unit (GPU) (Owens et al., 2008), training processes of large models have been accelerated heavily. That is because GPUs have a unique advantage for matrix multiplication, which usually requires a cubic time algebraic operation ($\mathcal{O}(N^3)$) and is the major and costly building block of NN computations. However, the number of concurrent operations required in large matrix multiplications for classification with extensive number of classes has reached a limit for further speedups even using GPUs.

### 1.1 Negative Sampling

The common approach to address this challenge is known as negative sampling (Pennington et al., 2014; Jean et al., 2014; Rawat et al., 2019; Mikolov et al., 2013b). In Negative Sampling, we only sample a small subset of classes for each input and compute the softmax and cross-entropy function. This subset usually includes the positive (true) and a small set of negative (false) classes. Negative

sampling scales down the computations in the most cumbersome last layer, thereby making training efficient.

However, approximating full-softmax with small sub-sample results in poor convergence if the negative samples are not chosen appropriately. For instance, let us take the example of a recommendation system (predicting products relevant to a query) with a large number of products. If the input query is 'Nike Running Shoes', the true loss concentrates on the specific small number of confusing ('hard') negative classes like 'Adidas Running Shoes'. Since the number of classes is huge, random sampling is unlikely to identify this hard negative class. Other heuristics like frequent class sampling as negative samples are also unlikely to find these hard negatives most of the time. Clearly, without discriminating between closely related negative samples, the classifier cannot achieve good accuracy. Our experiments on recommendations datasets clearly indicate this sub-optimality of current negative sampling heuristics.

If there exists a way to sample the subset of confusing classes from the skewed distribution, the training progress would be largely accelerated. However, as evident from the example, such ground-truth distribution depends on the input sample and current model parameters. Moreover, this distribution varies significantly as training progresses. Consider the same query 'Nike Running Shoes', initially, when the network has not learned anything and has random weights, all classes are equally confusing. Thus, uniform sampling is optimal initially as the network has just started to learn. As the training progresses, the network's belief starts getting more concentrated on a few classes; at this time, a negative sample of say 'baby toys' is not at all useful because the network has already learned to tell them apart. The sampling distribution keeps changing, often drastically, as the training progresses.

To the best of our knowledge, there does not exist any statistical sampling scheme and implementation for adaptive Negative Sampling, where the cost of maintaining and updating the distribution, per iteration, is $\mathcal{O}(1)$ (independent of the number of classes). This is because the input, current true class, and parameters update all the sampling weights in every iteration. It is widely assumed that there is no such sampling scheme, and hence several heuristic alternatives are proposed.

The first set of alternatives use a static distribution. The most popular ones, implemented in Tensor-Flow, assume a static distribution such as the distribution based on the frequency of classes. Uniform sampling is another popular choice.

Learning-based alternatives are also proposed (Bamler & Mandt, 2020), where a machine learning generator predicts (or generates) the negative samples. The sampler is solving the same hard problem, prediction over a large number of classes, as a sub-routine. Most importantly, since the sampling distribution for the same data point shifts drastically throughout training, ML models are likely to suffer.

Negative sampling alternatives try to balance the sampling cost with quality. So far, negative sampling methods, other than the ones based on static sampling, have failed to demonstrate any training time improvements over the optimized full softmax implementation over GPUs. Static sampling strategies are known to be fast but lead to poor accuracy. With current strategies, the cost of improving the quality with current alternatives does not seem worth it over the GPU acceleration of softmax.

In this paper, we change this. Our work provides a truly constant time adaptive sampling scheme utilizing the recent advances in Locality Sensitive Sampling (Charikar & Siminelakis, 2017; Spring & Shrivastava, 2017a). More impressively, we provide an efficient implementation of our proposal on CPU, which outperforms TensorFlow's implementation of softmax and other negative sampling strategies on some of the best available GPUs (V100) in terms of wall-clock training time.

**Summary of Contributions:**

1) We propose **two** efficient schemes for sampling 'hard' negatives where the negative sampling distribution provably adapts to changing parameters and the data instance. Furthermore, the sampling cost is provably constant (independent of the number of classes).

2) We show that our technique is not only provably adaptive but also practical. We provide an efficient CPU implementation, in C++, of our negative sampling approach. We demonstrate the effectiveness of a truly constant time negative sampler by showing that our implementation significantly outperforms standard TensorFlow on V100 GPU in-wall clock speed, while retaining the accuracy.

3) We provide a rigorous evaluation of our proposal with its efficient implementation against full softmax and popular approximations like sampled softmax, frequency-based sampled softmax, top-K activation softmax, differentiation softmax (D-Softmax), and Noise Contrastive Estimation(NCE). We report the time-wise and iteration-wise precision on large recommendation datasets like Amazon-670K, WikiLSH-325K, and popular natural language processing dataset Text8 corpus.

## 1.2 LSH BASED HASH TABLES

In this section, we briefly describe the recent development of using locality sensitive hashing for sampling and estimation (Chen et al., 2019b; Spring & Shrivastava, 2017a; Charikar & Siminelakis, 2017; Spring & Shrivastava, 2017b). Locality Sensitive Hashing (Indyk & Motwani, 1998; Indyk & Woodruff, 2006) is a widely used paradigm for large scale similarity search and nearest neighbor search. LSH is a family of hash functions with a unique property that vectors 'close' *wrt* some distance metric are more likely to have the same hash code as opposed to vectors that are 'far' from each other. Formally, one sufficient condition for a hash family $\mathcal{H}$ to be a LSH family is that the *collision probability $Pr_{\mathcal{H}}(h(x) = h(y))$* is a monotonically increasing function of the similarity:

$$Pr_{\mathcal{H}}(h(x) = h(y)) = f(Sim(x, y)), \tag{1}$$

where $f$ is a monotonically increasing function.

The idea is to use the hash value of $x$, i.e., $h(x)$, to generate key of $x$ in the hash table. We first initialize $L$ hash tables by constructing a meta-LSH hash function using $K$ independent hash functions for each of them. For details, see (Andoni & Indyk, 2004). There are three major steps:

**Pre-processing Phase:** Given a dataset of size $n$, we first insert all the data points into the hash tables using the meta-LSH formed by concatenating $K$ independent LSH hash functions. We only store the index/pointer of the data point in the hash tables instead of the entire vector. The cost of addition is $K \times L$ hash computations followed by $L$ insertions in the buckets.

**Query Phase:** During the query phase, we use the same meta-LSH hash to compute the hash codes for the query. Then we probe the corresponding bucket of each table and retrieve samples from it. The union of candidates from all hash tables constitute the samples for the particular query.

**Update Phase:** If an existing element in the database is updated, we can delete it from the hash table and re-add it. The cost is equivalent to twice the insertion cost of an element which is $2 \times K \times L$.

## 1.3 ADAPTIVE SAMPLING VIEW OF LSH

Denote $p_{qx}$ be the probability of retrieving $x$ from the datasets, when queried with a given query $q$. In (Indyk & Motwani, 1998), it was shown that for $(K, L)$ parametrized LSH algorithm the precise form of $p_{qx} = 1 - (1 - \alpha^K)^L$, where $\alpha$ is the collision probability of query $q$ and $x$ under the given LSH function, i.e. $\alpha = Pr_{\mathcal{H}}(h(x) = h(q))$. $p_{qx}$ is monotonic in $\alpha$ which is further monotonic in the similarity between query $q$ and the data element $x$. Note the similarity measure is dependent on the LSH function in use. See (Spring & Shrivastava, 2017a) for details.

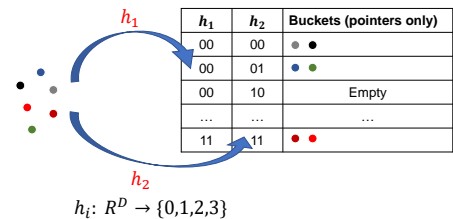

Figure 1: Schematic diagram of LSH. For an input, we compute hash codes and retrieve candidates from the corresponding buckets.

**Constant Time Sampling:** It should be noted that the cost of sampling is the cost of querying, which is only $K \times L$. This sampling cost is independent of the number of elements in the data. Clearly, the probability $p_{qx}$ is dependent on the query, and every element $x$ in the data has a different sampling probability. Thus, even though our sampling scheme induces $n$ different sampling probabilities every time the query $q$ is changed, the sampling cost is independent of $n$, and in-fact is constant if $K$ and $L$ are small constants. All this is assuming one $\mathcal{O}(n)$ time preprocessing.

This efficient sampling view of LSH has been used in a wide range of applications, such as deep neural networks (Spring & Shrivastava, 2017b; Chen et al., 2019a), kernel density estimation (Charikar & Siminelakis, 2017), record linkage (Chen et al., 2018), and optimization (Chen et al., 2019c).

Recent advances in fast inner product search using asymmetric LSH has made it possible to sample large inner products (Shrivastava & Li, 2014). Effectively, given a query $q$, it is possible to sample an element $x$ from the database with probability proportional to a monotonic function of inner product $f(q^T x)$. Here $f$ is a monotonically increasing function.

## 2 OUR PROPOSAL: LOCALITY SENSITIVE NEGATIVE SAMPLING (LNS)

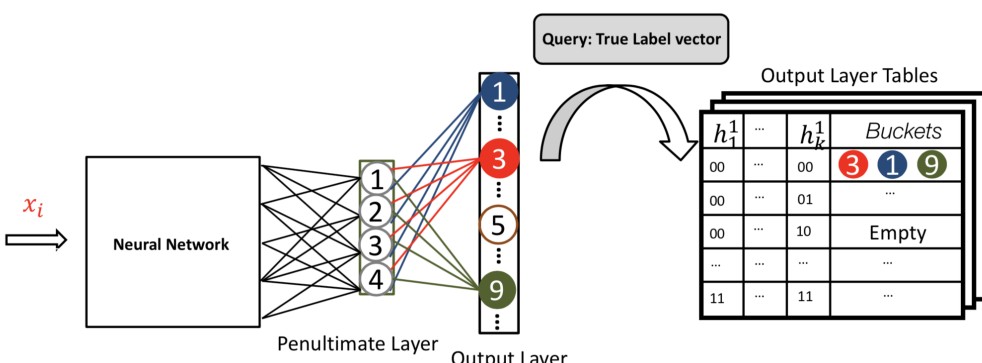

Figure 2: Schematic diagram of our proposal for LSH-label scheme. 1) We first construct hash tables for the label vectors. The label vectors are the weights of the connections from a label to the penultimate layer. In the figure, e.g. label vectors for node 1 (blue node) is a concatenation of its connection weights to the penultimate layer (blue lines). 2) For a training sample, we query the LSH tables with the true label weights (red lines) and obtain negative samples (blue and green nodes). We call the retrieved samples 'hard' negatives because they are very similar to the 'true' ones but are supposed to be 'false'.

**Notations:** We will start by defining a few vectors in the neural network setting illustrated in Figure 2. We are in large softmax settings. Here, we will use $N$ to denote the total number of classes. Define a vector $w_i \in \mathbb{R}^d$ (*class vectors*) to be the vector associated with class $i$ in the last layer of the neural network. We will use $(x, y)$ to denote the current input sample to the neural network for which we want to generate negative samples. We will use $E_x \in \mathbb{R}^d$ (*final input embedding*) to denote the vector of activation in the penultimate layer of the neural network when fed with input $x$.

We first describe our sampling procedure, and later we argue why it is distribution aware and constant time. Our approach, just like the LSH algorithm, has three phases. The first phase is a one time costly ($\mathcal{O}(N)$) prepossessing stage. The other two phases, sampling and update phase, are performed in each iteration, and both of them are constant-time operation independent of $N$.

**One time Preprocessing Phase during Initialization:** We start with randomly initializing the neural network parameters. This automatically initializes all the class vectors $w_i$. We now preprocess all these randomly initialized class vectors in $(K, L)$ parameterized LSH hash tables, as described in Section 1.2. This is a one-time operation during initialization.

**Sampling Phase for every input** $(x, y)$**:** In this phase, we process input $x$ to the penultimate layer and get the final input embedding $E_x$. Now instead of processing all the $N$ nodes in the last layer, we query the hash tables with either vector $E_x$ (**LSH Embedding**) or with the vector corresponding to the true label $y$, i.e., $w_y$ (**LSH Label**). This preciously describes our two sampling schemes. We can obviously mix and match, but we consider these two choices as two different methods for the simplicity of analysis and evaluations.

When we query, we generate a small set of the sampled candidates, call them $C$, forming our negative samples. Thus, we only compute the activation of nodes belonging to $C \cup y$ in the last layer and treat others as zero activation.

**Update Hash Tables with Update in Weights:** During backpropagation for input $(x, y)$, we only update $C \cup y$ weights in the last layer. We update these changed weights in the LSH hash tables.

Next, we first argue why this sampling is distribution aware and adaptive with every parameter and input change. We will then argue that the sampling and update process is significantly efficient. It is a constant-time operation that is easily parallelizable.

## 2.1 WHAT IS THE SAMPLING DISTRIBUTION? IS IT ADAPTIVE?

We start with two theorems that give the precise probability distribution of sampling a class as a negative sample with LSH Label and LSH Embedding methods provided the input $(x, y)$ and current parameters. We will use $p_{xy}$ as the collision probability of the LSH hash value of $x$ and $y$.

**Theorem 1 LSH Label Distribution** *For an input $(x, y)$ and LSH parameters $(K, L)$, the probability of sampling a class $i \neq y$ as negative sampling with LSH Label method is given by*

$$p_i \propto 1 - (1 - p_{w_y w_i}^K)^L,$$

*where $w_y$ and $w_i$ are the weights associated with true class $y$ and $i$ respectively. Furthermore, the probability of sampling class $i$ is more than any other class $j$, if and only if $sim(w_y, w_i) > sim(w_y, w_j)$. Here $sim$ is the underlying similarity function of the LSH.*

**Theorem 2 LSH Embedding Distribution** *For an input $(x, y)$ and LSH parameters $(K, L)$, the probability of sampling a class $i \neq y$ as negative sampling with LSH Embedding method is given by*

$$p_i \propto 1 - (1 - p_{E_x w_i}^K)^L,$$

*where $E_x$ is the embedding vector of input x and $w_i$ is the weights associated with class $i$ respectively. Furthermore, the probability of sampling class $i$ is more than any other class $j$, if and only if $sim(E_x, w_i) > sim(E_x, w_j)$. Here $sim$ is the underlying similarity function of the LSH.*

**Comments:** The expressions of probability are immediate from the sampling view of LSH. The expressions $1 - (1 - p^K)^L$ is monotonically increasing in $p$, the collision probability, which in turn is monotonically increasing in the underlying similarity function $sim$. Clearly, the distribution is adaptive as they change with the input $(x, y)$ as well as the parameters. So any update in the parameter or any change in the input change the sampling distribution completely. However, the sampling cost is constant and independent of the number of classes we are sampling from!

**Intuition of LSH Label:** Coming back to our example of class 'Nike Running Shoes'. Let us focus on LSH Label distribution. Initially, when all other labels have random weights, the similarity between the label 'Nike Running Shoes' and any other label will be random. So initial negative sampling should be like uniform sampling. However, as the learning progress, it is likely that 'Nike Running Shoes' and 'Adidas Running Shoes' will likely get close enough. Their weights will have high similarity (high $sim$), at that time, the LSH Label sampling will select 'Adidas Running Shoes' as likely negative sample for 'Nike Running Shoes' class.

**Intuition of LSH Embedding:** The LSH Embedding method is also adaptive. Consider the similarity function as an inner product. LSH embedding inner product with class vector is directly proportional to its activation. Thus, it naturally selects classes in which the classifier is confused (high activation but incorrect) as negative samples. Again, the distribution is adaptive.

## 2.2 COMPUTATIONAL COST FOR PROCESSING EACH INPUT

Given an input $(x, y)$, the cost of processing it without any negative sampling is $\mathcal{O}(N)$. With our proposed negative sampling the cost of sampling is the cost of query which is $K \times L$, a negligible number compared to $N$ in practice.

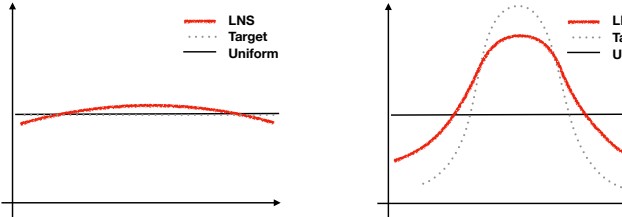

Figure 3: How the negative smapling distribution (target), uniform negative sampling and LNS adapts over iterations. Initially, when there is no learning all the sampling is close to uniform (Left Plots). During later states the sampling distribution is significantly different from uniform.

The cost of update is slightly more $(|C| + 1) \times K \times L$ because we have to update $|C| + 1$ weights. In negative sampling, $C$ is a very small constant. Also, in practice $K$ and $L$ are small constants. Furthermore, we have a choice to delay the hash table updates, as it only changes the sampling probability of few elements.

## 2.3 ALGORITHM AND IMPLEMENTATION DETAILS

First we construct $K \times L$ hash functions and initialize the weights of the network and $L$ hash tables. The LSH hash code of weight vectors of the last layer are computed and the id of the corresponding neuron is saved into the hash buckets. During the feed-forward path in the last layer, we query whether the embedding vector (LSH-embedding scheme) or the label vector of true class (LSH-label scheme) and retrieve the classes from hash table which are considered as negative classes. Instead of computing the activation of all the output nodes (full softmax), we compute the activations of the true classes and the corresponding retrieved negative classes. For the backpropagation, we backpropagate the errors to calculate the gradient and update the weights for the active nodes. Please refer to Algorithm 1 in the Appendix [A.4].

## 3 EXPERIMENTS

In this section, we will empirically evaluate the performance of our LSH Negative Sampling (LNS) approach against other sampling schemes that are conducive to GPUs. The real advantage of LNS is noticeable with huge neural networks. The popular extreme classification challenges have models with more than 100 million parameters, which are ideal for our purpose. For these challenges, most of the heavy computations happen in the last layer.

## 3.1 DATASETS

We evaluate our framework and other baselines on three datasets. Amazon-670K and WikiLSH-325K are two datasets from extreme classification repository (Bhatia et al., 2016) and Text8 is a popular NLP dataset. The detailed statistics about the dimensions and samples sizes are included in Table 1 in appendix [A.1].

## 3.2 BASELINES

We benchmark our proposed framework against full-softmax, sampled-softmax, topK-softmax, frequency-based-softmax, noise contrastive estimation and differentiated softmax (all explained below). All the baselines are implemented on TensorFlow-GPU. To have a fair comparison, the architecture, optimizer and size of hidden layer are exactly the same for all the methods on each dataset. Please note that our proposal only uses CPU and yet outperforms the other methods.

**Full-Softmax**: Full-softmax updates the weights of all the output neurons, which makes it computationally expensive and intractable for extreme classification framework. **Sampled-Softmax**: Sampled-softmax draws negative samples based on log-uniform distribution and updates their corresponding weights plus the weights for the true classes. This approach alleviates the computational

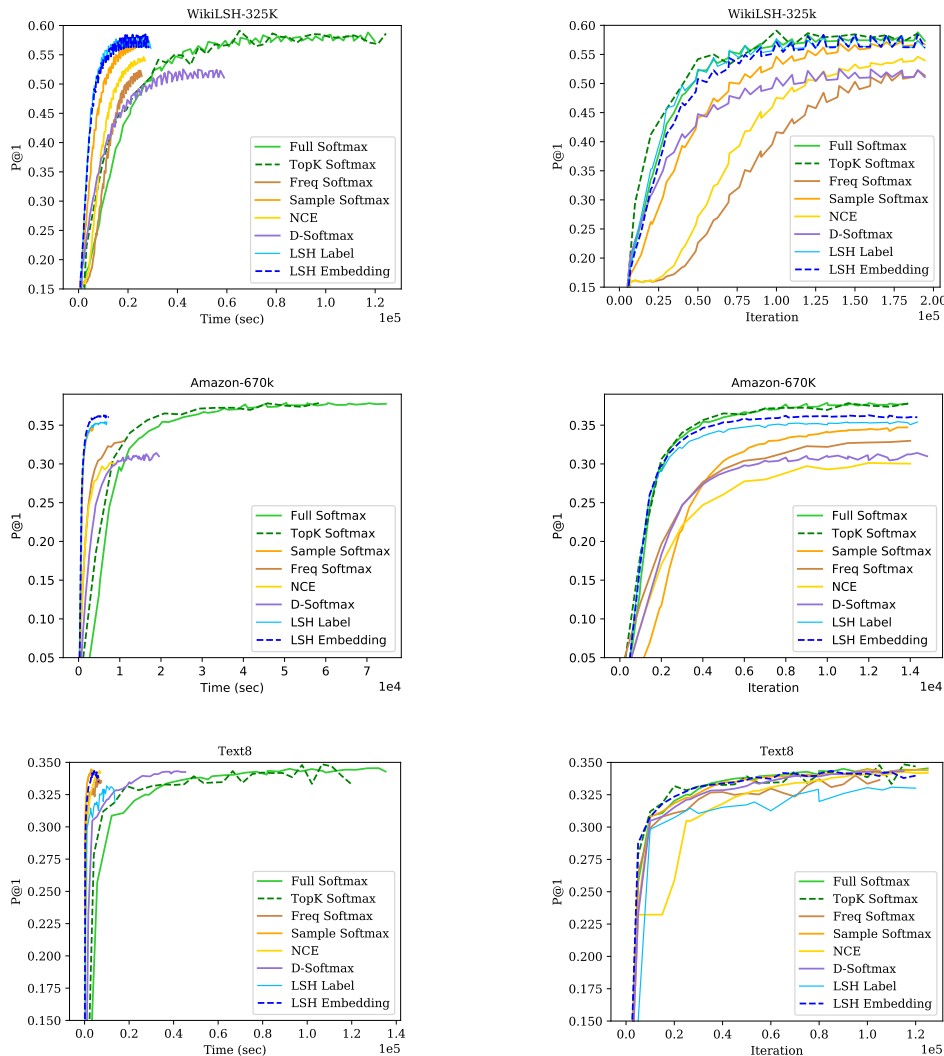

Figure 4: Comparison of our proposal LNS with two schemes (LSH label and LSH embedding) against full softmax, topK softmax, frequency based softmax, sampled softmax, NCE and D-softmax for all three datasets. **Left Column:** Precision@1 vs time, **Right Column:** Precision@1 vs iteration, **Top Row**: WikiLSH-325K dataset **Middle Row**: Amazon-670K dataset **Bottom Row**: Text8 dataset. The LSH methods closely mimics full softmax in iteration wise plots indicating the superiority of distribution aware sampling. The time plots clearly indicates the speed of sampling, where LSH sampling are the best performing ones.

bottleneck but degrades the performance in terms of accuracy. **TopK-Softmax**: TopK-softmax updates the weights of the output neurons with $k$ highest activations (including the true classes). This framework maintains better accuracy than sampled-softmax but with a slower convergence rate due to the scoring and sorting of all classes. **Frequency-based-Softmax:** Frequency-based-softmax samples the classes in proportion to the frequency of their occurrence in the training data. Computationally, this is the same as Sampled-softmax, however it samples negative classes from more frequent classes with higher probability. **Noise-Contrastive Estimation (NCE):** NCE loss (Gutmann & Hyvärinen, 2010) tackles multi-class classification problem as multiple binary classifiers instead. Each binary classifier is trained by logistic loss to distinguish between true classes and negative classes. Negative classes are sampled from a noise distribution which is typically log-uniform distribution or based on class frequencies. **Differentiated Softmax (D-Softmax):** D-softmax (Chen et al.,

2015) defines softmax layer weight matrix as a sparse block diagonal matrix where the blocks are constructed based on class frequencies, e.g. more frequent classes attains more parameters. **LNS (our proposal)**: Our proposed negative sampling algorithm samples the classes from output distribution which is adaptive to the input, true class and model parameters. Our model utilizes LSH to sample the most confusing (the most similar but false) classes as the negative samples in constant time.

### 3.3 ARCHITECTURE AND HYPERPARAMETERS

For Amazon-670K and WikiLSH-325K we use a standard fully connected neural network with hidden layer size of 128, where both the input and output are multi-hot encoded vectors. For Text8, We utilize standard word2vec language model with hidden layer size of 200, where input and output are one-hot and multi-hot encoded vectors, respectively. In word2vec architecture we utilize skip-gram model introduced in Mikolov et al. (2013a). Skip-gram model aims to predict nearby words in a document by learning continuous representation of words. Particularly, given a word, skip-gram model targets to predict the $m$ left and $m$ right neighbor words, where $m$ is window size and is considered as a hyperparameter. We pick $m = 2$ for our experiments.

We performed hyperparameter tuning for all the baselines to maintain their best trade-off between convergence time and accuracy. The optimizer is Adam with learning rate 0.0001 for all the experiments. The batch size for Amazon-670, WikiLSH-325 and Text8 is 1024, 256 and 512 respectively for all the experiments. We apply hash functions for the last layer where we have the computational bottleneck. In LSH literature, $L$ denotes the number of hash tables and $K$ denotes the number of bits in the hash code for each hash table (thereby having $2^K$ buckets per hash table). We use DWTA hash function [A.3] for Amazon-670K and WikiLSH-325K with K=6, L=400 and K=5, L=350 respectively. For Text8 we use Simhash hash function [A.2] with K=9 and L=50. We update the hash tables with an initial update period of 50 iterations and then exponentially decaying the updating frequency (as we need less updates near convergence). Our experiments are performed on a single machine with 28-core and 224-thread processors. All the baselines are run on the state of the art NVIDIA V100 GPUs with 32 GB memory.

### 3.4 RESULTS

Figure 4 shows the plots comparing $Precision@1$ (denoted here-on by P@1) versus both wall-clock time and the number of iterations for our method and all the baselines. For WikiLSH-325K dataset, LSH-label and LSH-embedding are respectively 4x and 4.3x faster than TensorFlow full softmax on GPU in terms of wall-clock training time. Moreover, both of them outperform all other TensorFlow baselines on GPU with significant margin. The same is true for Amazon-670K where LSH-label and LSH-embedding are 10x and 9.6x faster than TensorFlow full softmax on GPU correspondingly. On Text8 dataset, all methods perform more or less equally in terms of P@1 and our LSH-Embedding is the second fastest negative sampling method after sampled softmax in terms of training time. According to iteration-wise plots, both LSH-label and LSH-Embedding schemes attain the highest P@1 among all other sampling methods for Amazon-670K and WikiLSH-325K datasets, and achieves pretty similar performance with other sampling methods on Text8 dataset. Therefore, both variations of our LNS method outperform almost all other TensorFlow baselines on all datasets while being very similar to full softmax iteration-wise. This establishes the earlier statement that LNS does not compromise performance for speed-up. This is particularly noteworthy because our implementation of LNS uses only CPU while all other baselines run TensorFlow-GPU on a V100.

## 4 CONCLUSION

We proposed two provable, efficient and adaptive negative sampling schemes for neural networks with extreme number of classes. Our method samples negative classes in constant time, while adapts to the continuous change of the input, true class and network parameters. We efficiently implemented our algorithm on CPU in C++ and benchmarked it against standard TensorFlow implementation of six baselines on GPU. Our method on CPU outperforms almost all the TensorFlow baselines on GPU with significant margin on three datasets.

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

Table 1: Statistics of the datasets

|  | Feature Dim | Feature Sparsity | Label Dim | Train Size | Test Size |
|---|---|---|---|---|---|
| Amazon-670K | 135909 | 0.055 % | 670091 | 490449 | 153025 |
| WikiLSHTC-325K | 1617899 | 0.0026 % | 325056 | 1778351 | 587084 |
| Text8 | 253855 | 0.0004 % | 253855 | 13604165 | 3401042 |

# A  APPENDIX

## A.1  DATASET

**Amazon-670K** dataset is a product recommendation dataset with 670K labels. Here, each input is a vector representation of a product, and the corresponding labels are other products (among 670K choices) that a user might be interested in purchase. This is an anonymized and aggregated behavior data from Amazon and poses a significant challenge owing to a large number of classes.

**WikiLSHTC-325K** is based on two main sources: Wikipedia and ODP web directory. Each instance is in sparse vector format where the value of a feature corresponds to the frequency and the label corresponds to the category of the instance.

**Text8** is a popular NLP dataset and a preprocessed version of the first 100million tokens of English Wikipedia which contains 253K vocabulary. We utilized standard word2vec language model for text8 dataset, where input and output are one-hot and multi-hot encoded vectors, respectively.

## A.2  SIMHASH: SIGNED RANDOM PROJECTIONS

SimHash is a popular LSH that originates from *Signed Random Projections* (SRP) (Charikar, 2002; Rajaraman & Ullman, 2010; Henzinger, 2006) and preserves the cosine similarity measure. Given a vector $x$, SRP generates a random $w$ vector with each component generated from *i.i.d.* normal, , $w_i \sim N(0, 1)$, and only stores the sign of the projection. Formally SimHash is given by

$$h_w^{sign}(x) = sign(w^T x). \tag{2}$$

It was shown in (Goemans & Williamson, 1995) that the collision probability under SRP satisfies the following formula:

$$Pr(h_w^{sign}(x) = h_w^{sign}(y)) = 1 - \frac{\theta}{\pi}, \tag{3}$$

where $\theta = cos^{-1}\left(\frac{x^T y}{||x||_2 \cdot ||y||_2}\right)$.

## A.3  DWTA HASH: DENSIFIED WINNER TAKE ALL HASH

DWTA (Chen & Shrivastava, 2018) hash transforms the data into the transformed space such that their hamming distance correlates with their rank similarity measure in the original space. Densified WTA (DWTA) hash combines traditional WTA hashing and densification to improve discriminative power over sparse datasets. DWTA hash generates $\frac{KLm}{d}$ number of permutations and each permutation is split into $\frac{d}{m}$ bins where $d$ is the input dimension and $m << d$ is a hyperparameter. DWTA loops through the nonzero indices of the sparse input and updates the current maximum index of the corresponding bins according to the mapping in each permutation. It has been shown (Chen & Shrivastava, 2018) that the collision probability of DWTA is precisely the collision probability of WTA hash for nonempty bins, irrespective of the sparsity.

$$Pr(h_{Dwta}(x) = h_{Dwta}(y)) = Pr(h_{wta}(x) = h_{wta}(y)|\Theta(x) = \Theta(y) = Empty) \tag{4}$$

where $\Theta(x)$ is the set of random features of $x$ with permutation $\Theta$.

A.4   ALGORITHM

---

**Algorithm 1** Locality Sensitive Negative Sampling (LNS)

---

**input**  $E_x$ final input embedding, $w_i$ class vectors
**output**  $N_o$ set of active neurons of the last layer
  1: Initialize weights $w_l$ for last layer $l$
  2: Create $K \times L$ hash functions and initialize $L$ hash tables for the last layer
  3: Compute $h_l(w_l)$ for all output neurons
  4: **for** $i = 1 : Iterations$ **do**
  5:    **for** Batch B **do**
  6:      **if** LSH Embedding **then**
  7:        $N_o$=Query($h_l(E_x), HT_l$)
  8:      **end if**
  9:      **if** LSH Label **then**
10:        $N_o$ = Query($h_l(w_i), HT_l$)
11:      **end if**
12:    **end for**
13: **end for**

---

