# OpenReview forum: "A Truly Constant-time Distribution-aware Negative Sampling"
_ICLR.cc/2021/Conference — Reject_

### Official Review · AnonReviewer4 · 2020-10-27
**Simple yet effective approach for efficient negative sampling**

**Rating:** 5
**Confidence:** 4

**Review:**

When the number of classes is very large, calculating softmax for classification (e.g., in backpropagation) is computationally costly. Approaches based on negative sampling have been used in literature to alleviate this problem. However, most of existing approaches are (argued to be) either inaccurate or computationally costly. This paper proposes to use the well-known LSH (locality sensitive hashing) method to address this problem. In particular, two variants, LSH label and LSH Embedding are showed to speed up the training in terms of time needed to converge compared with a number of baseline methods over three large scale datasets.

+ The suggested approach is simple, making it potentially useful in practice.
+ The methodological contribution of the paper is more or less an off-the-shelf use of LSH for negative sampling. This being said, the application of LSH in this context is (seemingly) new, and the two basic similarity measures are interesting.
+ I would be cautious about calling the method "A truly constant-time" method. If we assume K and L are constant then yes; but theoretically in order to get good result we need to have larger values especially for L. Please elaborate on this.
+ The two theorems offered in the paper are the theorems from the LSH literature. Since a proxy is being used for similarity (e.g., label or embedding) these do not translate into the final classification result. The authors can perhaps elaborate on this and also give clues on how large K and L should be set depending on the parameters of the dataset, etc.
+ In intro we have "We show that our technique is not only provable but .."; it is not clear what is exactly provable, and how it relates to the classification result.

---

> ### Author Response · Authors · 2020-11-20
> **Response to Reviewer4**
>
> We thank the reviewer for their valuable comments.
>
> - The total cost of sampling is KL hash computations, followed by a constant number  (< 100) of bucket lookups. The KL hash computations can be done with fast hash functions in one read of the vector, so essentially the hashing cost is the same as input vector reading cost. The cost of aggregating a constant number (At most L) buckets is negligible compared to the cost of sorting N numbers (where N is in hundreds of thousands).  If this was not the case there is no way we could have achieved faster training time on CPU compared to Tensorflow packages which use the powerful V100.
> In theory, the cost is O(KL) which is independent of L in practice. K is smaller than 10 and L is in few hundreds, so the computation is in few thousands, compared to hundreds of thousands which is the size of N.
> - Please refer to the “intuition of LSH Label/Embedding” and the theorems in section 2.1 of the paper. As stated in the paper, a desirable negative sampling should be dependent on the input and the parameters (which keep changing over time). When the network has not learned anything (initial phases of training), any random class is equally confusing, and hence random sampling is desirable. However, near convergence, as mentioned in the paper, the best negative class is the closest enough class (example stated in the paper is "Adidas shoes" vs. "Nike shoes"). No method in the literature has this adaptivity as the training progresses. Either they are static (sampled softmax) or only dependent on input (learning-based). However, none of them will change the distribution as the network learns. Our method is the first to have this adaptivity where the sampling cost is O(1). The collision probability (p) in theorem1 and theorem2 is with respect to the last layer's parameters, which will be random in the initial phases. Hence, our samples will be similar to random samples. However, as the network learns and figures out how to discriminate between different categories, similar classes based on last layers’ weights are selected more frequently.
> -Results are sensitive to K due to the exponential growth of buckets, and the rule of thumb is K={8,9} for simhash hash function and K={5,6} for DWTA hash function. The results are not sensitive to choice of L. We only tried a few choices of K in all the experiments and it is mentioned in the paper.
> - Our technique is provably adaptive. We have modified this part in the paper.

---

### Official Review · AnonReviewer1 · 2020-10-28
**Looks like a great paper, somewhat sloppy writing**

**Rating:** 7
**Confidence:** 2

**Review:**

The paper studies extreme classification, where the number of classes is very large.

In that case, previous work came up with a technique to subsample bad classes for each data point to approximate cross-entropy or other used loss. Prior work used uniform samples, but this paper proposes an approach based on locality-sensitive hashing. One particular challenge that the authors need to overcome is updating LSH tables during the backpropagation (where parameters get updated).

The experiments are very convincing and show that CPU C++ implementation that uses new sampling can outperform GPU-based approach based on uniform sampling.

One comment is that the writing is fairly sloppy and could use additional polishing.

---

> ### Author Response · Authors · 2020-11-20
> **Response to Reviewer1**
>
> We thank the reviewer for the recognition of our work’s subtlety and appreciate their constructive comment. We have polished the paper and uploaded the new version.

---

### Official Review · AnonReviewer2 · 2020-11-01
**I think this paper over-claim its contribution even though it proposes a tractable solution to extreme classification.**

**Rating:** 3
**Confidence:** 4

**Review:**

This paper proposes a neighbor search method for negative sampling in extreme classification. The core idea is to use hash tables to select neighbors of a query.

I agree with the authors that:
i) The proposed hashing method can quickly retrieve neighbors for arbitrary queries in nearly constant time.
ii) Selecting hash functions is important for potential gain.

However, I think there are still missing pieces in this paper that need to clarify:
i) How do you calibrate the negative sampling results?
ii) How do you tune K and L in your experiments. To my understanding, using either too large or too small K and L is suboptimal in your setup.
iii) You mentioned that you are using CPU or computation. But the CPU and GPU precisions may contribute to the final result. Have you studied that in your experiments?

There are still some points that I disagree with:
i) There is no consensus that whether “easy negatives” or “hard negatives” should be selected in extreme classifications. For a recent survey on this topic, see “Embedding-based retrieval in Facebook search” by J. Huang et al. So I wonder if the hashing algorithm has the capacity to choose both easy and hard negatives?

ii) You claim you are using an O(1) algorithm independent of negative classes N. But this is simply not true and not reasonable because your K and L may need to increase when N grows in order to keep a constant collision probability.

iii) You mentioned you propose two schemes: “LSH Embedding” and “LSH Label”. But those two methods only differ in query representations. It would be interesting to compare these two carefully in experiments.

Overall, I think this paper over-claim its contribution even though it proposes a tractable solution to extreme classification.

---

> ### Author Response · Authors · 2020-11-20
> **Clarification on some misunderstandings**
>
> We thank the reviewer for their valuable comments.
>
> We would like to clarify the misunderstanding here. Our algorithm is not about near neighbor search, it is about sampling via LSH to retrieve the most confusing labels for the classifier. There is no notion of neighbor here, it is just similarity and sampling probability. We request reviewer to please revisit Theorem 1 and 2 for the insight. LSH is known to be expensive for exact search because it requires a large number of hash tables. If we use LSH for search, it will be orders of magnitude slower. We instead use LSH for adaptive sampling, which is very efficient and sub-linear because retrieving only a few buckets suffices for adaptive sampling.
>
> i) Most negative sampling is concerned with efficiency and does not require calibration. We followed the standard strategy in the Mikolov’s paper[1]
>
> [1]: Mikolov, Tomas, et al. "Distributed representations of words and phrases and their compositionality." Advances in neural information processing systems. 2013.
>
> ii) Results are sensitive to K due to the exponential growth of buckets, and the rule of thumb is K={8,9} for simhash hash function and K={5,6} for DWTA hash function. The results are not sensitive to choice of L. We only tried a few choices of K in all the experiments and it is mentioned in the paper.
> iii) We are not doing any precision tuning. The experiments with CPU use only CPU and experiments with GPU use standard Tensorflow packages.
>
> i) The aforementioned paper by  J. Huang et al. explores embedding-based search retrieval which is a different objective from our proposal. We propose a negative sampling framework which aims to approximate full softmax via label sampling. Hard negative sampling has been always known to be a significant approach for efficient training[1,2]. [3] argues that “hard” negatives maintain a better signal-to-ratio in gradient and encourage efficient training, while “easy negatives” retain lower signal-to-noise ratio due to the low correlation between input feature and sampled labels.
>
> [1]: Shrivastava, Abhinav, Abhinav Gupta, and Ross Girshick. "Training region-based object detectors with online hard example mining." Proceedings of the IEEE conference on computer vision and pattern recognition. 2016.\
> [2]: P. F. Felzenszwalb, R. B. Girshick, D. McAllester and D. Ramanan, "Object Detection with Discriminatively Trained Part-Based Models," in IEEE Transactions on Pattern Analysis and Machine Intelligence, vol. 32, no. 9, pp. 1627-1645, Sept. 2010, doi: 10.1109/TPAMI.2009.167.\
> [3] Bamler, Robert, and Stephan Mandt. "Extreme Classification via Adversarial Softmax Approximation." arXiv preprint arXiv:2002.06298 (ICLR 2020).
>
> ii) The total cost of sampling is KL hash computations, followed by a constant number  (< 100) of bucket lookups. The KL hash computations with fast hash functions can be done in one read of the vector so essentially the hashing cost is the same as input vector reading cost. The cost of aggregating a constant number (At most L) buckets is negligible compared to the cost of sorting N numbers (where N is in hundred of thousands.).  If this was not the case there is no way we could have achieved faster training time on CPU compared to all Tensorflow packages which use the powerful V100.
>
> In theory, the cost is O(KL) which is independent of L in practice. K is smaller than 10 and L is in few hundreds, so the computation is in few thousands, compared to the size of N.
>
> iii) We have shown the complete accuracy climb of the two methods with both iteration and running time. Yes, they seem to work similarly on most datasets except text8.
>
> We hope our clarifications on the misunderstandings will be taken into consideration and  our proposal method's subtlety is appreciated.

---

### Official Review · AnonReviewer3 · 2020-11-04
**LSH for negative sampling, as heuristic for top-k softmax. Implementation on multi-threaded CPU. Does not beat sampled softmax on GPU, gives no accuracy or approximation guarantees.**

**Rating:** 4
**Confidence:** 4

**Review:**

### Summary
The paper proposes a heuristic version of top-k negative sampling which is computationally effective. The main toolbox for this heuristic is locality sensitive hashing. The contribution is mainly algorithmic with an implementation. Experimental results support the effectiveness of the heuristic.

### Strengths
+ The use of LSH in machine learning applications is very promising, and this paper takes this trend to yet another direction where it can make a difference.
+ The approach is simple to describe and implement.

### Weaknesses
- There are several claims that are mostly marketing, namely “adaptivity” and “distribution awareness”. Yes, the sampling depends on the updated weights. But exactly how and in what manner that relates to, say, approximating the true objective, are not treated. The best intuition we can get is by thinking of the new method as a heuristic to top-k negative sampling, leading to believe that maybe some of the latter’s statistical/optimization properties are inherited. But this relationship is not made at all.
- The Theorems are mostly ornamental, they do not add anything new or relevant.
- It’s true that the proposed algorithm is amenable to a CPU implementation. But the CPU on which the experiments are run is a behemoth (28 core, 224 thread). So it clearly still needs the added parallelism to be competitive.
- The significance of the result are weakened when compared against one of the simplest alternatives, the sample softmax (orange curves in Figure 4). It is evident that in terms of reaching the vicinity of the ultimate accuracy, the sample softmax takes the same (if not less) time than the new method [the per-iteration plots are not very relevant, they demonstrate interim accuracy but have no bearing on ultimate accuracy and speed up].
So claims like “[these alternatives] fail to demonstrate any training time improvements” (page 2) are clearly false. Granted, sample softmax would be of order $O(\log N)$, but in practice this clearly is not a handicap.
- ~~For an application such as this, the details of LSH’s choices are critical, yet the paper only glosses over them. For example, the similarity metrics should differ between the “embedding” version and the “label” version, this is briefly alluded to in the experiments section without much explanation or references.~~ [Edit: thanks for clarifying this.]

### Overall
The motivation and approach of the paper are very strong. But the results are not as strong as they are hyped up to be, since a simple alternative achieves the same practical accuracy/time benefits and no guarantees are given for the accuracy of the new heuristic sampling, not even in terms of approximation of another heuristic such as top-k. This means any potential adoption would be based merely on experimental evidence. The fact that the algorithm can be implemented on a (highly parallel) CPU is not a good enough selling point, especially when not pitted against an equivalent optimized CPU implementation of the simple alternative.

_[Edit: The authors do not give any substantive feedback to my review, except for clarifying the hash choices. It is surprising that they object so vehemently to my intuitive description of their method as a heuristic to top-k, when they themselves write "Our proposed negative sampling scheme is a proxy to topK-softmax. It selects the top-k classes via LSH [...]". Also my reading of the sampled softmax is directly from their paper, showing a comparable accuracy-time tradeoff, but I was not refuted on this and instead was given other references claiming the inferiority of that method. I have updated my recommendation to reflect these shortcomings.]_

---

> ### Author Response · Authors · 2020-11-20
> **Not a heuristic for top-k softamx, and outperforms sampled softmax on GPU**
>
> We thank the reviewer for their valuable comments.
> - The exact objective for "negative sampling" is still a matter of debate. For instance, even the original Mikolov's paper [1] does not define any. However, as stated in the paper, a desirable negative sampling should be input dependent and dependent on the parameters (which keep changing over time). If we take the active sampling perspective [2,3], the most confusing class is an excellent negative sample. When the network has not learned anything (initial phases of training), any random class is equally confusing, and hence random sampling is desirable. However, near convergence, as mentioned in the paper, the best negative class is the closest enough class (example stated in the paper is "Adidas shoes" vs. "Nike shoes"). No method in the literature has this adaptivity as the training progresses. Either they are static (sampled softmax) or only dependent on input (learning-based). However, none of them will change the distribution as the network learns. Our method is provably, the first to have this adaptivity and where the sampling cost is O(1). If you look at Theorem 1 and Theorem 2, and the associated comments written in the paper, the collision probability (p) in those formulas is with respect to the last layer's parameters, which will be random in the initial phases. Hence, our samples will be similar to random samples. However, as the network learns and figures out how to discriminate between different categories, similar classes based on last layers’ weights are selected more frequently.  The proposal is NOT similar to the top-k heuristic because it rarely samples the top-k. All it guarantees is that the most similar element has the highest chance of sampling in expectations. It is also not heuristic because we can exactly compute the sampling probabilities.\
>  We can write a discussion on this if needed, but we hope our proposal method's subtlety is appreciated. It is easy to miss that because there is nothing in the literature capable of achieving that in constant time.
> - Please see previous comments. The theorems give the precise sampling probability and are needed to prove that they are adaptive. Also, the formulas provide insight into how the sampling relates with the learned weights, and how the  change of weights over time changes the negative sampling distribution.
> - Yes, our algorithm is data parallel and can utilize any number of cores, however taking into consideration that CPUs are much more economically viable than GPUs and moreover, they offer more flexibility in terms of memory, our algorithm amenability to CPUs is an advantage.
> - We believe that the reviewer’s statement “does not beat sampled softmax on GPU” is not correct. Sampled softmax in general is a fast proxy to full softmax, however it is not a desirable proxy due to its low accuracy. Please look at [4,5,6,7],  sampled softmax has poor accuracy and convergence even though it is fast. Same is observed in our experiments. For example on Amazon670 sampled softmax barely reaches 34% while ours easily goes to 36%.  So the efficiency comes at the cost of accuracy with static sampling like sampled softmax (as explained  in the paper).
> - In our experiments the hash function utilized for each dataset is the same for both versions of LSH Labl and LSH Embedding. We used the standard Signed Random Projections and DTWA for the hash functions and have added a part for them in the appendix (A.2 and A.3).  Choice of LSH is an interesting discussion which is left to future work, given that LSH is a more than 2 decade old topic and there are a variety of options.
>
> [1]: Mikolov, Tomas, et al. "Distributed representations of words and phrases and their compositionality." Advances in neural information processing systems. 2013.\
> [2]: Shrivastava, Abhinav, Abhinav Gupta, and Ross Girshick. "Training region-based object detectors with online hard example mining." Proceedings of the IEEE conference on computer vision and pattern recognition. 2016.\
> [3]: P. F. Felzenszwalb, R. B. Girshick, D. McAllester and D. Ramanan, "Object Detection with Discriminatively Trained Part-Based Models," in IEEE Transactions on Pattern Analysis and Machine Intelligence, vol. 32, no. 9, pp. 1627-1645, Sept. 2010, doi: 10.1109/TPAMI.2009.167.\
> [4] Rawat, Ankit Singh, et al. "Sampled softmax with random fourier features." Advances in Neural Information Processing Systems. 2019.\
> [5]Blanc, G. & Rendle, S.. (2018). Adaptive Sampled Softmax with Kernel Based Sampling.Proceedings of the 35th International Conference on Machine Learning, in PMLR80:590-599\
> [6]Bamler, Robert, and Stephan Mandt. "Extreme Classification via Adversarial Softmax Approximation." arXiv preprint arXiv:2002.06298 (ICLR 2020).\
> [7]Y. Bengio and J. -S. Senecal. 2008. Adaptive Importance Sampling to Accelerate Training of a Neural Probabilistic Language Model. Trans. Neur. Netw. 19, 4 (April 2008)

---

### Decision · Program_Chairs · 2021-01-07
**Final Decision**

**Decision:**

Reject

**Comment:**

All reviewers agree that this paper is interesting, but needs improvement in order to be suitable for a highly competitive venue such as ICLR. Reviewer 3 is especially incisive and detailed, but other reviewers make similar points.